# Acquired hypothyroidism, iodine status and hearing impairment in adults: A pilot study

**Tereza Grimmichova**[1,2]*, **Ludmila Verespejova**[3], **Zuzana Urbaniova**[3], **Martin Chovanec**[3], **Martin Hill**[1], **Radovan Bilek**[1]

1 Institute of Endocrinology, Prague, Czech Republic, 2 Department of Internal medicine, Faculty Hospital Kralovske Vinohrady and Third Faculty of Medicine, Charles University, Prague, Czech Republic, 3 Department of Otorhinolaryngology, Faculty Hospital Kralovske Vinohrady and Third Faculty of Medicine, Charles University, Prague, Czech Republic

* tgrimmichova@endo.cz

## Abstract

### Objectives

Hearing impairment can have major impacts on behavior, educational attainment, social status, and quality of life. In congenital hypothyroidism, the incidence of hearing impairment reaches 35–50%, while in acquired hypothyroidism there is a reported incidence of 25%. Despite this, knowledge of the pathogenesis, incidence and severity of hearing impairment remains greatly lacking. The aim of our study was to evaluate hearing in patients with acquired hypothyroidism.

### Methods

30 patients with untreated and newly diagnosed peripheral hypothyroidism (H) and a control group of 30 healthy probands (C) were enrolled in the study. Biochemical markers were measured, including median iodine urine concentrations (IUC) µg/L. The hearing examination included a subjective complaint assessment, otomicroscopy, tympanometry, transitory otoacoustic emission (TOAE), tone audiometry, and brainstem auditory evoked potential (BERA) examinations. The Mann-Whitney U test, Fisher's Exact test and multivariate regression were used for statistical analysis.

### Results

The H and C groups had significantly different thyroid hormone levels (medians with 95% CI) TSH mU/L 13.3 (8.1, 19.3) vs. 1.97 (1.21, 2.25) p = 0 and fT4 pmol/L 10.4 (9.51, 11.1) vs. 15 (13.8, 16.7) p = 0. The groups did not significantly differ in age 39 (34, 43) vs. 41 (36,44) p = 0.767 and IUC 142 (113, 159) vs. 123 (101, 157) p = 0.814. None of the hearing examinations showed differences between the H and C groups: otomicroscopy (p = 1), tympanometry (p = 1), TOAE (p = 1), audiometry (p = 0.179), and BERA (p = 0.505).

### Conclusions

We did not observe any hearing impairment in adults with acquired hypothyroidism, and there were no associations found between hearing impairment and the severity of

**Data availability statement:** All relevant data are within the manuscript.

**Funding:** Ministry of Health, Czech Republic—conceptual development of a research organization (Institute of Endocrinology—EU, 00023761). The funders had no role in study design, data collection and analysis, decision to publish, or preparation of the manuscript.

**Competing interests:** The authors have declared that no competing interests exist.

hypothyroidism or iodine status. However, some forms of hearing impairment, mostly mild, were very common in both studied groups.

## Introduction

Hypothyroidism can occur in countries that have sufficient iodine supplementation, with autoimmune thyroiditis the most common cause [1]. Patients with hypothyroidism report hearing losses; however, hearing impairments are poorly recognized and underreported, and often explicitly associated with congenital hypothyroidism and iodine deficiency [2]. It is well known that hearing deficits have a great impact on the education skills and social status of children and adolescents [3]. Youth with hearing impairments experience behavioral problems and demonstrate lower performance in oral language compared with peers with normal hearing [4]. Hearing impairment is also associated with increased cognitive dysfunction and dementia in the elderly [5]. Unfortunately, only limited studies have been dedicated to acquired hypothyroidism and hearing disabilities in adults [2,6,7].

The development of ears and their functioning depends on thyroid hormones. Thyroid hormones are necessary for maturation of the cochlea and the central auditory areas [8]. In humans, the critical period for hearing maturation corresponds approximately to an interval ranging from the early embryonic period to the first year of postnatal life [2]. Hypothyroidism occurring during this critical time window can lead to irreversible hearing impairment [9,10]. However, when thyroxine (T4) therapy is begun early enough in life, it might still successfully reverse hearing loss [11]. Iodine is an essential component of the thyroid hormones triiodothyronine (T3) and thyroxine (T4). Thyroid hormone receptors (TR), occurring as the two isoforms TRα and TRβ, regulating thyroid hormone metabolism at the transcriptional level are expressed in the cochlea [12,13]. Genetic disorders such as Pendred syndrome [14], thyroid hormone resistance [15], and thyroid hormones monocarboxylate transporter 8 abnormalities [16] support the links between thyroid hormones and the auditory system.

Overall, it seems that hearing impairment associated with acquired hypothyroidism are most often bilateral, symmetrical, and sensorineural (perceptive), with some studies also supporting a conductive loss, and reversible to varying degrees after levothyroxine replacement therapy [6,17]. Bircher was the first to describe the association of hearing impairment in patients suffering from a goiter as early as 1883 [18]. Various investigators have reported an association between thyroid hormone concentrations and hearing function in humans, but many of these studies are from the last century. In the 1950s, a study by Howarth and Lloyd 1956 described deafness under hypothyroidism as perceptive in type, with hypertrophy and edema of the mucosa of the nose, middle ear and eustachian tubes causing eustachian obstruction and thickening of the tympanic membrane. The study included seven females aged from 31–75 years with various levels of conductive hearing impairments (slight 0-20 dB loss to very severe perceptive deafness over 60 dB loss) and demonstrated improvements in some patients after thyroid hormone substitution [19]. In the 1970s the study Bhatia et al. 1977 confirmed hypothyroidism using estimates of serum protein-bound iodine in 43% of patients (total 72 patients) who had mild hearing loss as assessed by pure tone audiometry [20]. In the 1980s, twenty hypothyroid patients were investigated, demonstrating hearing losses in 80% of patients when compared with randomly selected age- and sex-matched normal subjects. Following treatment with levothyroxine a statistically significant improvement in hearing thresholds was observed by pure-tone audiometry. The investigators suggested a causal relationship between hypothyroidism and hearing loss [6]. In 2002, Malik et al. also reported a higher degree of hearing impairments in forty-five hypothyroid patients (age range 10–57 years). Pure tone

audiometry (PTA) revealed hearing impairments in thirty-two patients, out of which fifteen had conductive, nine had mixed and eight had sensorineural impairments. Hearing impairment was correlated with increasing serum TSH levels and decreasing serum T3 and T4 levels (p > 0.05). Brainstem Evoked Response Audiometry (BERA) was performed in patients having hearing impairments shown by PTA. The interpeak interval (I-V) was > 4.0 ms in 81.80% of ears, and the waves were not well formed with lower amplitude. After treatment with levothyroxine, hearing thresholds were significantly better in 30% of ears, with conductive impairment more likely to be improved [7]. The vestibular system was found to be affected only minimally [20]. Similar results but to a lesser extent were shown in a study by Aggarwal et al., with only about 13% cases having an objective audiological improvement, though many cases claimed a subjective improvement in hearing after thyroid hormone substitution. One case also had a deterioration in hearing [21]. In contrast, some studies have shown no hearing impairment in hypothyroid patients [22,23] and others have not supported associations between hearing impairment and thyroid hormone levels in hypothyroid patients [24,25].

Further, the evidence does suggest that iodine deficiency is related to hearing loss, and that supplementation in iodine-deficient individuals may improve hearing thresholds. Auditory impairment due to hypothyroidism causing such an iodine deficiency might exist [3]. The National Health and Nutrition Examination Survey is a cross-sectional, nationally representative survey of the noninstitutionalized civilian population of the United States analyzing data from 1198 adolescent (aged 12–19 years) participants. Urinary iodine concentration (UIC) less than 100 μg/L was found to be a predictive risk factor for having speech-frequency hearing loss among adolescents and more specifically among those with UICs less than 50 μg/L. However, the geometric mean of thyrotropin concentration of 1.4 μUI/mL was in the normal range [26,27]. In adults however, even less is about acquired hypothyroidism in relation to auditory function.

The aims of our study were to assess:

1) Hearing impairment in relation to acquired hypothyroidism and associated changes to thyroid hormone levels.

2) Hearing impairment due to iodine status as measured by median urinary iodine concentrations.

3) The risks of hearing loss and associations with the severity of hypothyroidism, including whether or not the severity of deafness is proportional to the degree of hypothyroidism and/or urinary iodine concentrations.

## Patients and methods

The protocol of this prospective study complied with the Declaration of Helsinki and was approved by the Ethic Committee of the Faculty Hospital Kralovske Vinohrady. Before entering the study, written informed consent was obtained by the investigator from patients after they received both written and oral information. Patients' history, medical records, ultrasound of the neck (US), and biochemical testing were done at two workplaces: the Institute of Endocrinology and the Department of Internal Medicine of the University Hospital Kralovske Vinohrady in Prague. Hearing tests were done by a trained audiologist at the Department of Otorhinolaryngology of the University Hospital Kralovske Vinohrady in Prague. A total of 30 hypothyroid patients and 30 matched healthy probands without history/treatment of thyroid disease were enrolled in the study. The patients were all from the Czech Republic, a country with iodine sufficiency [28], and were consecutively recruited from 25th May 2021 to 21st February 2024. The inclusion and exclusion criteria for control and hypothyroid groups

are given in the study flow chart in Fig 1. Briefly, we excluded any congenital and acquired hearing disorders including systemic diseases impairing hearing or ototoxic medications.

Overt primary hypothyroidism is defined as thyroid-stimulating hormone (TSH) concentrations above the reference range and free thyroxine (T4) concentrations below the reference range. The diagnosis of overt hypothyroidism was confirmed by a second blood sample [29]. Both the control and hypothyroid groups of patients underwent the same study protocol with the same examinations. Biochemical testing consisted of measurement of electrolytes, liver and kidney tests, blood count, lipids, glucose, and Hb1ac to exclude diabetes, dyslipidemia, and systemic disease with impact on hear loss. Three consecutive morning urine samples were taken to measure median urinary iodine concentrations (UICs). UIC is an indicator of iodine status and reflects the current dietary intake of iodine. Urinary iodine below 20 μg/L denotes severe iodine deficiency, between 20–49 moderate, between 50–99 mild iodine deficiency, UIC between 100–199 is adequate iodine intake, UIC between 200–299 more than adequate, and UIC more than 300 μg/L is excessive iodine intake [30]. The WHOQOL-BREF questionnaire was used to evaluate the quality of life [31]. The hearing examination consisted of descriptions of subjective symptoms (tinnitus, dizziness, feeling of hear loss, and balance problem) and objective hearing exams: tympanometry (Maico MI 24), pure tone audiometry (Clinical Audiometer AC 40 Interacoustic), transitory otoacoustic emission (TOAE) (Titan Interacoustic) and Brain Evoked Response Audiometry (BERA) (Eclipse Interacoustic). Hearing losses were classified as either: mild (20–34 dB); moderate (35–49 dB); moderately severe (50–64 dB), severe (65–79 dB), profound (80–94 dB), or as complete hearing loss ( > 95 dB).

**Fig 1. Flow chart of the study.**

The Global Burden of Disease (GBD) scale defines hearing loss as the quietest sound an individual can hear in their better ear, taken as the pure-tone average (PTA) of audiometric thresholds of 0.5 kHz, 1 kHz, 2 kHz, and 4 kHz [32,33]. Serum TSH (0.270–4.200 mIU/L), fT4 (11.9–21.6 pmol/L), fT3 (3.10–6.80 pmol/L), TRAbs (0.00–1.75 IU/L), thyroglobulin (3.50–77.00 ug/L) concentrations were measured using the ECLIA method (Roche). The HbA1C test was performed using an ion exchange HPLC method that is certified by the NGSP (www.njsp.org) and standardized or traceable to the Diabetes Control and Complications Trial (DCCT) reference assay. Serum anti-Tg (0.01–120.00 IU/mL) and anti-TPO (0.01–40.00 IU/mL) were measured by ELISA (Aeskulisa). Urinary iodine concentrations were measured by absorption spectrophotometry at a wavelength of 430 nm. Glucose was measured by a spectrophotometry (UV)-hexokinase method. Liver enzymes and lipids were measured by absorption spectrophotometry (Cobas PRO, Roche), and blood count (Celtac F, Nihon Kohden).

## Statistics

Statistical significance was defined as p-values < 0.05. The Mann-Whitney U test (medians with 95% CI) and Fisher's Exact test were used for statistical analysis (Number Cruncher Statistical Systems (NCSS) software, Kaysville, UT, USA). The importance of individual predictors in discriminating between individual groups was evaluated using multivariate regression with a reduction of dimensionality known as orthogonal projections to latent structure (OPLS) for one predicted (dependent) variable in the model (SIMCA v. 12.0 statistical software, UMETRICS, Umeå, Sweden) [34]. The effect size for discrimination between groups for significance level $p < 0.05$ and power = 0.8 was estimated using Mann Whitney test. In our data we were able to differentiate between the groups with effect size 0.385 (to find a moderate effect). The power analysis was performed using PASS 2023 NCSS software, Kaysville, UT, USA.

## Results

A total of 60 patients were enrolled in the study, but two patients had to be excluded: one hypothyroid patient due to newly diagnosed severe iron deficiency anemia and one control proband due to newly diagnosed polycythemia vera. Finally, 29 hypothyroid patients (H; 21 women and 8 men) and 29 controls (C; 25 women and 4 men) were statistically analyzed. The H and C groups were not statistically different in either age (medians with 95% CI) 39 (34, 43) vs. 41 (36, 44) (p = 0.767) or gender (p = 0.331). The H group had significantly higher levels than the C group of: TSH 13.3 (8.1, 19.3) vs. 1.97 (1.21, 2.25) mU/L (p = 0); anti-TPO 335 (164, 520) vs. 28 (4.78, 31) IU/mL (p < 0.01); and anti-Tgl IU/mL 32.6 (4.88, 58.6) vs. 1.3 (1.3, 1.3) IU/mL (p < 0.001). The H group had significantly lower fT4 10.4 (9.51, 11.1) vs. 15 (13.8, 16.7) pmol/L (p = 0) and fT3 4.61 (4.35, 5.1) vs. 5.24 (5.07, 5.5) pmol/L (p = 0.009) compared to the C group. For more details see Table 1.

The cause of hypothyroidism was autoimmune thyroiditis (p = 0). The H and C groups were comparable in IUCs 142 (113, 159) vs. 123 (101, 157) μg/L (p = 0.814). Only 15 probands had mild iodine deficiencies, with the lowest IUC of 50 μg/L. For more details see Table 2.

Out of the 58 patients in the study group, only 3 patients in the C group presented with subjective complaints, including unilateral hearing loss and tinnitus. There were no subjective complaints in the H group (p = 0.239). Otomicroscopy revealed no pathologies except for one case of an atrophic eardrum (p = 1). Some hearing impairment in the patients was observed in 29 patients, of which 17 were in the hypothyroid group and 12 control probands. Hearing impairments predominantly affected high-tone frequencies (6 kHz and 8 kHZ) in the range from 20-45dB. We did not observe any deficiency of more than 45 dB. The hearing examinations tympanometry (p = 1), TOAE (p = 1), audiometry (p = 0.179), and BERA (p = 0.505) were not different between the H and C groups. For more details see Tables 2 and 3.

**Table 1. The Mann-Whitney test was used to compare the control and hypothyroid groups.**

| Variable | Control group (median 95% CI) | Hypothyroid group (median 95% CI) | p-value |
|---|---|---|---|
| Age | 41 (36, 44) | 39 (34, 43) | 0.767 |
| BMI kg/m² | 23 (21.6, 23.9) | 25.6 (24, 28.3) | 0.08 |
| WHOQOL-BREF | 97 (94,102) | 95 (91, 104) | 0.63 |
| TSH mIU/L | 1.97 (1.21, 2.25) | 13.3 (8.1, 19.3) | 0 |
| anti-TPO IU/mL | 28 (4.78, 31) | 335 (164, 520) | <0.001 |
| anti-Tgl IU/mL | 1.3 (1.3, 1.3) | 32.6 (4.88, 58.6) | <0.001 |
| Thyroglobulin µg/L | 12.7 (8.85, 24.5) | 8.29 (1.34, 26.4) | 0.217 |
| UIC µg/L | 123 (101, 157) | 142 (113, 159) | 0.814 |
| Thyroid volume mL | 9.38 (8.6, 10.5) | 11.9 (10, 13.2) | 0.019 |
| Total cholesterol mmol/L | 4.67 (4.47, 4.84) | 5.51 (5.29, 6.11) | <0.001 |
| LDL cholesterol mmol/L | 2.84 (2.38, 3.1) | 3.46 (2.95, 3.83) | 0.003 |
| HDL cholesterol mmol/L | 1.6 (1.55, 1.69) | 1.49 (1.26, 1.8) | 0.643 |
| Triglycerides mmol/L | 1.07 (0.85, 1.17) | 1.37 (1.04, 1.69) | 0.009 |
| Glucose mmol/L | 5.1 (4.68, 5.33) | 5.01 (4.7, 5.52) | 1 |
| HbA1c mmol/mol | 33.5 (32, 35) | 33 (32, 36) | 1 |

WHOQOL-BREF, a questionnaire for quality of life; TRAK, thyroid-stimulating hormone receptor antibodies; IUC, iodine urine concentration.

**Table 2. Fisher's exact 2-sided test was used to compare the control and hypothyroid groups.**

| Variable | Control group | Hypothyroid group | p-value |
|---|---|---|---|
| Male | 13.8% | 27.6% | 0.331 |
| Smoking | 3.7% | 18.2% | 0.160 |
| Autoimmune thyroiditis | 3.4% | 86.2% | 0 |
| Otomicroscopy | 0% | 3.4% | 1 |
| Tympanometry | 6.9% | 6.9% | 1 |
| Audiometry | 27.6% | 10.3% | 0.179 |
| GBD R | 3.4% | 0% | 1 |
| GBD L | 3.4% | 0% | 1 |
| BERA | 13.8% | 21.4% | 0.505 |

GBD R, GBD scale right ear; GBD L, GBD scale left ear; BERA, brainstem evoked response audiometry.

**Table 3. The Mann-Whitney test was used to compare the control and hypothyroid groups.**

| Variable | Control group (median 95% CI) | Hypothyroid group (median 95% CI) | p-value |
|---|---|---|---|
| PTA R (dB) | 9 (8, 10) | 8 (6, 9) | 0.120 |
| PTA L (dB) | 9 (8, 10) | 9 (6, 10) | 0.456 |
| Audiometry 6 kHz R (dB) | 10 (10, 15) | 10 (10, 15) | 0.833 |
| Audiometry 6 kHz L (dB) | 10 (10, 20) | 15 (10, 15) | 0.768 |
| Audiometry 8kHz R (dB) | 10 (10, 15) | 15 (10, 20) | 0.387 |
| Audiometry 8kHz L (dB) | 10 (10, 20) | 15 (10, 20) | 0.880 |

PTA R, pure-tone average right ear; PTA L, pure-tone average left ear; R, right ear; L, left ear.

The OPLS statistical analysis used to evaluate the importance of individual predictors to discriminate between the hypothyroid and control groups found no statistically significant predictors of hearing impairments.

## Discussion

In our study, we did not observe any hearing impairments related to acquired hypothyroidism in adults. Further our data did not support any associations between hearing impairment and the degree of hypothyroidism and iodine status. However, some forms of hearing impairment, mostly mild, were very common in both studied groups.

The prevalence of overt hypothyroidism in the general population varies between 0–2% and 3-5% in Europe, and the prevalence of undiagnosed hypothyroidism, including both overt and mild cases, is around 5%. In iodine-sufficient areas, the most common cause of hypothyroidism is chronic autoimmune thyroiditis [29]. We concur with these results and we have to point out that we had some difficulties finding a patient with manifest hypothyroidism. Only three patients had TSH levels over 50 mU/L. The probable reason was that our patient cohorts were younger, but widespread thyroid function testing and relatively low thresholds for treatment initiation almost certainly also play a role [35].

Hearing impairment is a major public health problem and is one of the leading causes of years lived with a disability. There are many reasons for hearing impairment in adults and it is strongly associated with age. We chose to study younger probands (<50 years old) in order to eliminate the risks of presbycusis, metabolic disorders such as type 2 diabetes, obesity, hypertension and dyslipidemia with their associated negative effects on hearing [36–38]. Not surprisingly, our hypothyroid group of probands had higher lipids levels compared to the control group, but without any impact on hearing impairment. The majority of probands were non-smokers, so this risk factor was not relevant in this study. In comparison to some other studies [6,7,20] we can just speculate on the divergent results. Most of the studies were done in the past, and recent studies are mostly from India and the Middle East [24,39,40]. This geographical variation could introduce several confounding factors, including the prevalence of occupational noise exposure, preventable infections such as chronic otitis media and meningitis, nutritional status and health-care access. Further, there is evidence of improvements in hearing between 1959–1962 and 1999–2004, suggesting a beneficial trend for at least half a century. Explanations for this trend could include reductions in occupational noise exposure, less smoking, and better management of other cardiovascular risk factors such as hypertension and diabetes [32].

Previous studies were also frequently done with a small number of probands with ages ranging from 12 to 75 years old. Further, the hypothyroid status of probands tended to be very heterogeneous, from subclinical forms to severe myxedema [41]. In the study of Malik et al., 2002, moderate hearing impairment was present approximately in 48% and mild impairment in 41% of ears, whereas severe or profound hearing impairment was not found in any case [7]. In agreement with previous studies, we observed mostly mild and only two cases with moderate hearing impairments in both hypothyroid and control groups. In later studies, hearing examinations were usually more detailed than thyroid testing and evaluations of iodine status [7,24,39]. Serum thyroglobulin is well correlated with the severity of iodine deficiency as measured by median urinary iodine concentrations (UICs) [42], but our groups had comparable levels of both thyroglobulin and UICs in the optimal range. Based on WHO data, the number of countries with iodine insufficiency has declined from 113 (estimated by total goiter rates) in 1993 to 20 (estimated by urinary iodine concentrations) in 2017 [43]. Congenital iodine deficiency can be manifested as two syndromes: a more common neurological disorder with brain damage, deaf mutism, squint and spastic paresis of the legs. Such patients are usually

euthyroid, but goiter and hypothyroidism can be seen in some cases. Urinary iodine levels are usually less than 20 μg/L. The less common syndrome includes severe hypothyroidism and growth retardation but a less severe mental defect. Both conditions are due to dietary iodine deficiency and can be prevented by iodine supplementation before pregnancy [44]. Hearing loss has also been observed in adolescents (aged 12–19 years) associated with iodine deficiency in spite of normal TSH levels [26]. Therefore, it remains uncertain whether the hearing loss is solely caused by hypothyroidism or iodine deficiency. It appears that iodine plays a role in immune responses and might have a beneficial influence on mammary dysplasia and fibrocystic breast disease [45]. It is possible that iodine itself could have some other roles in human physiology.

One of the limits of our study is the low number of probands, but this was rather a pilot study about a topic with very limited evidence. We also did not retest the patients after thyroxine replacement therapy reaching euthyroidism, since there were no significant differences between the hypothyroid and control groups in hearing impairments. There are some previous studies showing significant improvement of hearing threshold in 12.5 up to 73% ears at several levels, middle ear, cochlear or retro-cochlear [39]. Bhatia et al 1979 reported definite subjective improvement in hearing on becoming euthyroid, but this was not confirmed by audiometry. Another limit of our study could be that the hypothyroid state was not too advanced in our patients. All of our hypothyroid patients complied with the definition of hypothyroidism, i.e. increased levels of TSH and simultaneously decreased fT4 levels in spite of fT3 being in normal range in some of the probands. There are adaptive mechanisms during iodine deficiency, or the incipient phase of hypothyroidism, preserving optimal plasma and tissue T3 levels linked to deiodinase 2 expression. The signs and symptoms of overt hypothyroidism are minimized by preserving T3 levels. Thus, serum T3 levels perform poorly as a tool to diagnose hypothyroidism compared to serum FT4 or TSH. Nevertheless, it is clear that plasma and tissue T3 contents are associated and that changes in plasma T3 are directly connected to changes in the tissue T3 content [46]. We cannot exclude that patients with parallel drops in T4 and T3 levels could be at risk of hearing impairments, but our statistical analysis did not observe any links between thyroid hormone levels and such impairment. A benefit of our study is the prospective design with comprehensive examinations of patients and excluding more common causes of hearing loss. We included evaluations of iodine status, which has a crucial effect on hearing impairment. Finally, the quality of life of our probands established by the WHOQOL-BREF questionnaire was comparable for both groups.

## Conclusions

We did not observe any hearing impairment in adults associated with acquired hypothyroidism. However, some forms of hearing impairment, mostly mild, were very common in both studied groups. Since our study did not show any associations between hearing impairment and either the severity of hypothyroidism or iodine status, we suppose that there are more important causes of hearing impairment than hypothyroidism in adults.

## Author contributions

**Conceptualization:** Tereza Grimmichova.

**Data curation:** Tereza Grimmichova.

**Funding acquisition:** Tereza Grimmichova.

**Investigation:** Ludmila Verespejova, Zuzana Urbaniova.

**Software:** Martin Hill.

**Supervision:** Martin Chovanec.

**Writing – original draft:** Tereza Grimmichova.

**Writing – review & editing:** Radovan Bilek.

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
