## [Decision Letter · Decision Letter 0]

11 Nov 2024

PONE-D-24-20543Acquired hypothyroidism, iodine status and hearing impairment in adults: a pilot study.PLOS ONE

Dear Dr. Grimmichova,

Thank you for submitting your manuscript to PLOS ONE. After careful consideration, we feel that it has merit but does not fully meet PLOS ONE’s publication criteria as it currently stands. Therefore, we invite you to submit a revised version of the manuscript that addresses the points raised during the review process.

<please by="" manuscript="" revised="" submit="" your="">plosone@plos.org. Please include the following items when submitting your revised manuscript:</please>

Please submit your manuscript by Dec 26 2024 11:59PM. If applicable, we recommend that you deposit your laboratory protocols in protocols.io to enhance the reproducibility of your results. Protocols.io assigns your protocol its own identifier (DOI) so that it can be cited independently in the future. For instructions see: https://journals.plos.org/plosone/s/submission-guidelines#loc-laboratory-protocols . Additionally, PLOS ONE offers an option for publishing peer-reviewed Lab Protocol articles, which describe protocols hosted on protocols.io. Read more information on sharing protocols at https://plos.org/protocols?utm_medium=editorial-email&utm_source=authorletters&utm_campaign=protocols .

We look forward to receiving your revised manuscript.

Kind regards,

Preeti Kanawjia, MD

Academic Editor

PLOS ONE

Journal Requirements:

2. Please remove all personal information, ensure that the data shared are in accordance with participant consent, and re-upload a fully anonymized data set. 

3. Thank you for stating the following financial disclosure: Ministry of Health, Czech Republic—conceptual development of a research organization (Institute of Endocrinology—EU, 00023761).

Reviewers' comments:

Reviewer's Responses to Questions

**Comments to the Author**

1. Is the manuscript technically sound, and do the data support the conclusions?

Reviewer #1: No

Reviewer #2: Yes

2. Has the statistical analysis been performed appropriately and rigorously? 

Reviewer #1: No

Reviewer #2: Yes

3. Have the authors made all data underlying the findings in their manuscript fully available?

Reviewer #1: Yes

Reviewer #2: Yes

4. Is the manuscript presented in an intelligible fashion and written in standard English?

Reviewer #1: Yes

Reviewer #2: Yes

5. Review Comments to the Author

Reviewer #1: Study is underpowered with only 30 cases/ controls and cannot answer the research question that hypothyroid patients are more prone to hearing loss than euthyroid controls ( given the anticipated effect size and precision of measurement).

The introduction and discussion are excessively detailed. The results section similarly has extensive information not relevant to the research question.

Reviewer #2: The manuscript “Acquired hypothyroidism, iodine status and hearing impairment in adults: a pilot study investigate the rate of hearing impairment in acquired hypothyroid patients. Manuscript designed well and consider for publication. However there are few corrections, which need to incorporate in manuscript.

1. Page 6, line no. 148, explain the criteria for selection of controls and mention that Hearing acuity test was performed in control group or not.

2. Page number 7, line number 159, 160 and 161, the authors have mentioned that Overt primary hypothyroidism is defined as thyroid-stimulating hormone (TSH) concentrations above the reference range and free thyroxine (T4) concentrations below the reference range. The diagnosis of overt hypothyroidism was confirmed by a second blood sample. The authors need to mention reference range and diagnosis criteria with appropriate use of reference.

3. Page 8, line no. 189, mention the name of statistical software used for data analysis.

4. Line no. 146-147, Hearing tests were done at the Department of Otorhinolaryngology of the University Hospital Kralovske Vinohrady in Prague. The hearing examination was conducted by trained audiologist or not?

6. PLOS authors have the option to publish the peer review history of their article (what does this mean? ). If published, this will include your full peer review and any attached files.

**Do you want your identity to be public for this peer review?** For information about this choice, including consent withdrawal, please see our Privacy Policy .

Reviewer #1: No

Reviewer #2: **Yes: ** Dr. PAWAN KUMAR KARE

---

## [Author Response · Author response to Decision Letter 0]

27 Nov 2024

Response to Reviewers

First of all, we would like to thank the reviewers for reading and making the comments to our work. We appreciate it very much.

Reviewers' comments #1: Study is underpowered with only 30 cases/ controls and cannot answer the research question that hypothyroid patients are more prone to hearing loss than euthyroid controls ( given the anticipated effect size and precision of measurement).

Response to Reviewer #1: We did the power analysis. The effect size for discrimination between groups for significance level p<0.05 and power=0.8 was estimated using Mann Whitney test. In our data we were able to differentiate between the groups with effect size 0.385 (to find a moderate effect). We added to our statistics.

Reviewers' comments #1: The introduction and discussion are excessively detailed. The results section similarly has extensive information not relevant to the research question.

Response to Reviewer #1: We removed some of the excessively detailed parts from the introduction, results and discussion. You are right, we were too excited about too many details. However, we wanted to show the topic of hearing impairment and hypothyroidism comprehensively. There are too many unknown gaps.

Reviewers' comments #2:

Page 6, line no. 148, explain the criteria for selection of controls and mention whether the hearing acuity test was performed in the control group or not.

Response to Reviewer #2: We explained it in more detail and hope to make it clearer (marked).

Reviewers' comments #2: Page number 7, line number 159, 160 and 161, the authors have mentioned that Overt primary hypothyroidism is defined as thyroid-stimulating hormone (TSH) concentrations above the reference range and free thyroxine (T4) concentrations below the reference range. The diagnosis of overt hypothyroidism was confirmed by a second blood sample. The authors need to mention reference range and diagnosis criteria with appropriate use of reference.

Response to Reviewer #2: The reference ranges of our lab are mentioned below (marked), we used the appropriate reference.

Reviewers' comments #2: Page 8, line no. 189, mention the name of statistical software used for data analysis.

Response to Reviewer #2: We added all the softwares.

Reviewers' comments #2: Line no. 146-147, Hearing tests were done at the Department of Otorhinolaryngology of the University Hospital Kralovske Vinohrady in Prague. The hearing examination was conducted by trained audiologist or not?

Response to Reviewer #2: Yes, by a trained audiologist (marked). ________________________________________

---

## [Decision Letter · Decision Letter 1]

8 Jan 2025

Acquired hypothyroidism, iodine status and hearing impairment in adults: a pilot study.

PONE-D-24-20543R1

Dear Dr. Grimmichova,

We’re pleased to inform you that your manuscript has been judged scientifically suitable for publication and will be formally accepted for publication once it meets all outstanding technical requirements.

Kind regards,

Preeti Kanawjia, MD

Academic Editor

PLOS ONE

Additional Editor Comments : "I have considered all reviewer comments and feel that the authors have provided sufficient explanation to support an Accept decision despite the alternative recommendation from a reviewer.")

Reviewers' comments:

Reviewer's Responses to Questions

**Comments to the Author**

1. If the authors have adequately addressed your comments raised in a previous round of review and you feel that this manuscript is now acceptable for publication, you may indicate that here to bypass the “Comments to the Author” section, enter your conflict of interest statement in the “Confidential to Editor” section, and submit your "Accept" recommendation.

Reviewer #1: (No Response)

Reviewer #2: All comments have been addressed

2. Is the manuscript technically sound, and do the data support the conclusions?

Reviewer #1: No

Reviewer #2: Yes

3. Has the statistical analysis been performed appropriately and rigorously? 

Reviewer #1: I Don't Know

Reviewer #2: Yes

4. Have the authors made all data underlying the findings in their manuscript fully available?

Reviewer #1: Yes

Reviewer #2: Yes

5. Is the manuscript presented in an intelligible fashion and written in standard English?

Reviewer #1: Yes

Reviewer #2: Yes

6. Review Comments to the Author

Reviewer #1: (No Response)

Reviewer #2: The manuscript “Acquired hypothyroidism, iodine status and hearing impairment in adults: a pilot study" investigate the rate of hearing impairment in acquired hypothyroid patients. Manuscript designed well and consider for publication. The authors responded appropriately and incorporated all the information in the revised manuscript.

7. PLOS authors have the option to publish the peer review history of their article (what does this mean? ). If published, this will include your full peer review and any attached files.

**Do you want your identity to be public for this peer review?** For information about this choice, including consent withdrawal, please see our Privacy Policy .

Reviewer #1: No

Reviewer #2: **Yes: ** Dr. PAWAN KUMAR KARE

---

## [Editor Report · Acceptance letter]

PONE-D-24-20543R1

PLOS ONE

Dear Dr. Grimmichova,

I'm pleased to inform you that your manuscript has been deemed suitable for publication in PLOS ONE. Congratulations! Your manuscript is now being handed over to our production team.

Kind regards,

on behalf of

Dr. Preeti Kanawjia

Academic Editor

PLOS ONE